# Asymptomatic Stroke in the Setting of Percutaneous Non-Coronary Intervention Procedures

**DOI:** 10.3390/medicina58010045

**Published:** 2021-12-28

**Authors:** Giovanni Ciccarelli, Francesca Renon, Renato Bianchi, Donato Tartaglione, Maurizio Cappelli Bigazzi, Francesco Loffredo, Paolo Golino, Giovanni Cimmino

**Affiliations:** 1Vanvitelli Cardiology Unit, Monaldi Hospital, 80131 Naples, Italy; frances.renon@gmail.com (F.R.); rmbianchi@hotmail.it (R.B.); d.tartaglione@virgilio.it (D.T.); mcappellibigazzi@gmail.com (M.C.B.); francesco.loffredo@unicampania.it (F.L.); paolo.golino@unicampania.it (P.G.); giovanni.cimmino@unicampania.it (G.C.); 2Department of Translational Medical Sciences, Section of Cardiology, University of Campania “Luigi Vanvitelli”, 80131 Naples, Italy; 3Molecular Cardiology, International Centre for Genetic Engineering and Biotechnology, 34149 Trieste, Italy

**Keywords:** silent cerebral ischemia, asymptomatic stroke, non-coronary cardiac procedures

## Abstract

Advancements in clinical management, pharmacological therapy and interventional procedures have strongly improved the survival rate for cardiovascular diseases (CVDs). Nevertheless, the patients affected by CVDs are more often elderly and present several comorbidities such as atrial fibrillation, valvular heart disease, heart failure, and chronic coronary syndrome. Standard treatments are frequently not available for “frail patients”, in particular due to high surgical risk or drug interaction. In the past decades, novel less-invasive procedures such as transcatheter aortic valve implantation (TAVI), MitraClip or left atrial appendage occlusion have been proposed to treat CVD patients who are not candidates for standard procedures. These procedures have been confirmed to be effective and safe compared to conventional surgery, and symptomatic thromboembolic stroke represents a rare complication. However, while the peri-procedural risk of symptomatic stroke is low, several studies highlight the presence of a high number of silent ischemic brain lesions occurring mainly in areas with a low clinical impact. The silent brain damage could cause neuropsychological deficits or worse, a preexisting dementia, suggesting the need to systematically evaluate the impact of these procedures on neurological function.

## 1. Introduction

Cardiac catheterization is associated with a low incidence of thromboembolic stroke. In particular, symptomatic events are described from 0.1 to 0.6% during diagnostic coronary catheterization [1,2], while this is lower related to percutaneous coronary intervention (PCI) [3]. The higher incidence is reported in studies that showed systematic brain magnetic resonance imaging (MRI) and brain infarcts detected by diffusion-weighted MRI (DW-MRI) during invasive cardiac procedures can be present in 8% of patients [4]. However, the high number of procedures performed every year worldwide and the morbidity rate increase the risk of debilitating effects related to stroke [5,6].

At the same time, the periprocedural stroke in the setting of non-coronary percutaneous intervention is mostly due to the embolization from the aortic arch to the brain artery of atherosclerotic and calcified material mobilized by coronary catheters or dedicated devices during the procedures [7,8]. Interventions for valvular and structural heart diseases are associated with a significant risk of embolic complications [9]. Any left-sided structural heart intervention, such as manipulation of calcified and degenerated valves, records an incidence of symptomatic stroke from 2 to 7%, according to the procedures [10,11].

Silent cerebral infarcts (SBIs) are clinically silent brain lesions diagnosed by DW-MRI, and they are associated with a higher incidence of dementia and cognitive impairment [12,13,14]. They are observed in the majority of the patients undergoing transcatheter procedures, and in up to 90% of those undergoing transcatheter aortic valve replacement (TAVR) [15,16].

The indications for percutaneous interventions for structural heart disease are continuously increasing [17,18], and the correlation with the peri-procedural risk of stroke is still unclear. The objective of this review is to evaluate the risk of cerebral complication linked to non-coronary transcatheter cardiac procedures, focusing on TAVR, percutaneous mitral valve repair and left atrial appendage occlusion (Figure 1).

## 2. Silent Ischemic Stroke and Cognitive Dysfunction

The key question of whether “silent” ischemic brain lesions are associated with neurological consequences is still unanswered. The prevalence of silent brain infarcts is increased in patients with a history of stroke and dementia. Although silent brain infarcts lack acute strokelike symptoms by definition, they are associated with subtle neurological deficits, often unnoticed by patients and physicians. In particular, silent brain infarcts have been associated with visual field deficits, limb disturbances, frailty, decline in physical function, worsening cognitive ability, vascular dementia and depressive symptoms [12,19,20,21].

In the Rotterdam Scan Study, a prospective cohort study conducted in patients free of dementia, the presence of silent brain infarcts at baseline doubled the risk of incident dementia at follow-up, with a worse cognitive impairment in subjects with multiple ischemic lesions. The decline in different cognitive domains varied with the location of silent brain infarcts on MRI [14].

Clinicopathological studies showed that patients with lacunar infarcts were more likely to have dementia and needed fewer tangles and plaques for a clinical diagnosis of Alzheimer’s disease [12].

In addition to macroscopic silent brain infarcts, smaller cerebral microinfarcts, detectable with high resolution structural MRI, have been associated with cognitive impairment and dementia [22].

In populations of atrial fibrillation patients, observational studies have shown association of anticoagulant treatment with a lower risk of incident dementia. A possible mechanism could be the prevention of cerebral microemboli and silent brain infarcts by anticoagulation [23]. Although this is indirect evidence, it supports the notion that microembolization may be a causal factor of cognitive decline and dementia.

Studies evaluating the association between DW-MRI lesions and neuropsychological deficits have led to controversial results: in patients undergoing coronary angiography or coronary artery bypass grafting, new DWI lesions were associated with a decline in neuropsychological test performance; this association was not found in other studies in patients after cardiac surgery. This discrepancy in results could be attributed to the small patient samples; however, it could also point to the possible role of ischemic lesions in cognitive decline in specific subgroups of patients [19].

A recent observational study showed newly acquired microembolic cerebral lesions in 9 out of 13 patients undergoing MitraClip implantation. Although all lesions resolved completely during follow-up, two patients with a higher number of new post-procedural lesions showed a significant decline in global cognitive function tests [24].

Further studies are needed to elucidate the neurological consequences of silent cerebral ischemic lesions.

## 3. Percutaneous Transcatheter Procedures and Risk of Cerebral Ischemia and Stroke

### 3.1. Transcatheter Aortic Valve Replacement (TAVR)

Degenerative aortic valve stenosis is the most frequent valvular disease in Western countries [25]. Transcatheter aortic valve replacement (TAVR) represents the gold standard treatment in selected high-risk patients and could be considered in intermediate-risk patients [26,27].

Despite improvements in materials and technique having significantly reduced the rate of stroke as compared to the first experiences [28], TAVR is still correlated to cerebral complications.

According to the Valve Academic Research Consortium (VARC) definitions, stroke is defined as the onset of new neurological deficits, focal or global, that last more than 24 h and are caused by embolic, hemorrhagic or ischemic events. Compared to baseline clinical condition, stroke can be classified as disabling or non-disabling as defined by the mRS score (modified Rankin scale-score ≥ 2 at 90 days; mRS score < 2 at 90 days) [29]. On the other hand, transient ischemic attack (TIA) indicates the new onset of neurological deficit that persists less than 24 h (generally 1–2 h) without evidence of organ damage assessed by neuroimaging methods [29].

Several studies have evaluated the risk of cerebral ischemia correlated to TAVR.

The Partner A and B trials included high-risk patients, showing no significant difference in 30-day stroke risk between TAVI and SAVR [30].

On the other hand, the PARTNER 3 trial enrolled low-risk patients highlighting a 75% reduction in stroke risk in the TAVI group compared to the SAVR group (0.6% vs. 2.4%, hazard ratio 0.25, 95% CI 0.07–0.88) [31].

However, data from meta-analysis [32,33,34] and registries [35] indicate a lower rate of stroke at 30 days and 1 year.

SBIs exhibit a higher incidence than clinical strokes in patients undergoing aortic valve replacement, and they are observed more frequently after TAVR than after SAVR (at least 70% [36] and 44% [37] respectively). The presence of multiple lesions and a bilateral distribution suggests an embolic etiology and particulates likely originate from the aortic valve and aorta [38].

Correlations have been noted between the number of new SBI lesions and both patient- and procedure-related factors. Diabetes mellitus and chronic kidney disease appear associated with a higher mean number of new SBIs: given their association with a higher cerebrovascular risk, their presence probably decreases the threshold required for a new ischemic event. Peripheral vascular disease, on the other hand, was not associated with a higher risk of new SBIs, potentially due to the lower likelihood of retrograde embolism of debris from peripheral disease. Amongst the procedure-related factors, the performance of an aortic valvuloplasty during the TAVR correlates with a higher number of new SBIs [36].

The temporary use of cerebral protection devices during TAVR procedures has demonstrated the presence of fragments of materials such as fibrin, amorphous calcified material and connective tissue derived from the flaps of the native aortic valve and atherothrombotic material from the aorta [39], able to determine the occlusion of the cerebro-afferent vessels in 75% of the patients subjected to transcatheter implantation.

Two mechanisms can be linked to TAVR-related stroke risk:

Periprocedural risk: the dilatation of the valve, the manipulation of the catheters and the activation of the inflammatory cascade can determine the exposure of thrombogenic factors. At the same time, the device itself can favor the formation of small thrombi [40,41];

Post-procedural risk: two factors can increase the possibility, over the year, to develop a stroke: valve thrombosis and new onset of atrial fibrillation (AF). Observational studies have demonstrated that the risk of cerebrovascular events is predominant in the first hours following the procedure, while subacute events (1 to 30 days after TAVI) were mainly associated with new onset of AF [42]. Late events (30 days to 1 year), were related to pre-existing AF, atherosclerosis and cardiovascular risk factors [42]. Hyperacute, subacute and chronic thrombosis in patients with normal trans-prosthetic gradients (mean gradient < 20 mmHg) have also been reported [43]. MRI can highlight the presence of silent strokes in almost 2/3 of TAVI patients, in some cases leading to cognitive dysfunction [15,44,45]. Long-term outcome of silent brain embolization remains unknown, and use of cerebral protection should be better investigated.

### 3.2. Mitral Valve Repair and Replacement

Mitral regurgitation affects more than 10% of people above 75 years [46]. This disease represents the second most frequent indication for valve surgery [17] and, despite the lack of randomized clinical trials, it is widely accepted that, when feasible, surgical valve repair is the preferred treatment [17,18]. Percutaneous edge-to-edge mitral valve repair with a MitraClip device (Abbott, Menlo Park, CA, USA) is an alternative to surgery in patients with severe and moderate-to-severe mitral regurgitation in patients who cannot be candidates for cardiac surgery [17,18,47].

Despite the controversial results available to date on the efficacy of this procedure (transcatheter edge-to-edge repair seems to reduce mortality only in certain subgroups of patients) and the need for an appropriate selection of the patients, there is consensus on the safety and feasibility of this procedure. In the Endovascular Valve Edge-to-Edge Repair Study (EVEREST) II, transcatheter mitral repair with the MitraClip device was safer than surgery but was not as effective in reducing the severity of mitral regurgitation [47]. In the safety analysis of the MITRA-FR (Percutaneous Repair with the MitraClip Device for Severe Functional/Secondary Mitral Regurgitation) study the peri-procedural complications accounted for 14.6% of the patients, which included two cases (1.4%) of embolism, resulting in overt stroke. At 2 years follow-up, the rate of ischemic or hemorrhagic stroke was 4.6%, with all the events observed during the first 12 months [48]. In the device group of the COAPT (Cardiovascular Outcomes Assessment of the MitraClip Percutaneous Therapy for Heart Failure Patients with Functional Mitral Regurgitation) population the rate of stroke at 2 years was 4.4% (versus 5.1% in the control group) [49]. A recent meta-analysis confirmed a similar incidence of stroke in patients treated with transcatheter edge-to-edge mitral valve repair (TEER) and in patients in medical therapy; a trend towards a lower stroke rate has been observed with TEER versus surgical valve repair [50].

Overall, considering data from trials and observational studies, reported clinically overt stroke after MitraClip implantation occurs in a very small percentage of patients, ranging from 0.2 to 1.2% for in-hospital events and from 0.7 to 2.6% at 30 days [11,51,52,53,54]. Cases of left atrial and left ventricular peri-procedural thrombus formation have been reported [55,56]. While the risk of peri-procedural overt stroke is low, the risk of clinically silent cerebral microembolism appears to be significant, despite the paucity of data available at present. A silent cerebral infarction is defined as a new post-procedural diffusion-weighted magnetic resonance imaging (DW-MRI) lesion without focal neurological abnormality. In a recent study, Blazek et al. [57] enrolled 27 high-risk patients to assess the incidence and impact of both clinically apparent and silent cerebral ischemia using serial DW-MRI before and after MitraClip procedure. As for heart-team discussion, the study population was considered ineligible for heart surgery. Almost 67% of the patients had atrial fibrillation, and 18% of the patients had a story of previous stroke, reflecting a high grade of comorbidities. Comparison of pre- and post-interventional brain MRI showed new embolic lesions spread in both hemispheres, with a diffuse pattern, in 85% of the patients, without any clinically overt stroke. In the majority of cases, multiple lesions were observed in different neurovascular territories of both hemispheres, suggesting an embolic mechanism. Device time was the only independent predictor of new ischemic MRI lesions at multivariate analysis. This could be explained by the possible role of device time as a marker of the procedure complexity and of the degree of valvular or subvalvular damage. High BMI and the number of clips needed for valve repair were not significatively associated with new lesions, despite a trend towards a higher burden of embolism [57].

There was no significant decrease in the post procedural cognitive function assessed by the Montreal Cognitive Assessment (MoCA) score compared with baseline, although the presence of more than three lesions on DW-MRI and mitral valve calcifications on echocardiography were predictors of lower scores in univariate analysis. Despite these findings, only the pre-procedural MoCA score was significantly associated with a post-procedural worsening of cognitive function [57].

The use of cerebral embolic protection devices (CEP) during MitraClip procedure has provided some insights on the mechanisms and pathophysiology of procedure-related stroke.

A recent study by Frerker et al. [58] included 14 high-risk patients undergoing MitraClip with the dual filter Sentinel system (Claret Medical, Santa Rosa, CA, USA) in situ. Interestingly, debris was found in 100% of the 14 patients who underwent MitraClip. The microscopical analysis of the 28 (proximal and distal) filters showed that, in patients treated with 2 clips instead of 1 clip, embolic particles had higher diameter (maximum particle diameter 402 μm vs. 134 μm and the cumulative particle area was bigger (3.45 mm^2^ vs. 0.81 mm^2^). The most commonly identified debris materials were acute thrombus and small fragments of basophilic foreign material consistent with hydrogel, each found in 86% of the patients [11,58]. Valve tissue and/or superficial atrial wall tissue were found in 64.3% of the patients, followed by organizing thrombus (28% of the patients) and microparticles of myocardium (14.6%). Interestingly, calcium was not found on the dual filter CEP devices, according to the previous observation that mitral calcifications seem to have no correlation with the development of new ischemic lesions [57].

Apart from demonstrating the safety and feasibility of cerebral embolic protection devices during transcatheter mitral valve repair, the study opened up new unanswered questions. First, it is not clear if the risk of overt and subclinical stroke could be reduced by the use of CEP devices, and further studies are needed [59]. Second, the origin of the debris particles is still unclear. Acute thrombus formation could derive from the guidewire system or the trans-septal sheat, as well as from thrombogenic surfaces of the clip delivery system. Interestingly, a rate of 9% of thrombus detection on the transseptal sheat despite optimal anticoagulation was reported [60]. The other major components of the embolic debris were small fragments of hydrogel. This polymeric coat is usually found on sheath and catheters, where it contributes to reducing mechanical friction with the vessel walls. Hydrogel embolism has been previously described in different intravascular procedures, where it appeared associated with multisystemic embolism [61,62,63].

While organizing thrombus embolization during MitraClip procedure seems to be related to the high prevalence of atrial fibrillation in the study population, the presence of fibroelastic tissue consistent with atrial or valve tissue could be attributed both to the transseptal puncture and to a procedural injury. Not surprisingly, it is associated with the delivery of multiple clips and with a longer device time, which reflects numerous grasping attempts [58].

A subsequent work tried to identify which TEER procedural step was associated with an increased risk of cerebral embolization by monitoring the procedure with continuous transcranial Doppler (TCD) [64]. Microembolization signals (MES) were identified in all examined patients, and they occurred predominantly during device interaction with the mitral valve. This association is supported by the aforementioned study by Frerker et al., in which valve and atrial wall tissue fragments represented the 64% of the debris remaining on the cerebral protection system after the procedure [58]. The number of MES was also associated with procedure length and with the implantation of two or more clips. A subgroup of patients in the same study underwent MRI examinations before and after the procedure. In 87% of these subjects, new DWI lesions occurred, in a pattern consistent with an embolic mechanism.

Apart from MitraClip, recent interest has been raised about transcatheter mitral valve replacement procedures (TMVR), and new devices have been designed for both native and prosthetic valves [65,66,67,68].

Although the feasibility of these interventional procedures has been demonstrated, most of the clinical studies are in very early stage, and no valid conclusion can be drawn on the clinical efficacy and the stroke risk of these procedures. In a recent study enrolling 64 high-risk patients undergoing TMVR with compassionate use of balloon expandable valves, the 30-day burden of clinically relevant stroke was 6.9% [67], but this high percentage has to be interpreted in the light of a high rate of comorbidities in the study population. Mitral valve calcifications have been addressed as the potential source of embolic particles [69]. The relative risk of stroke is almost double in patients with mitral annular calcifications (relative risk 2.10 (95 percent confidence interval, 1.24 to 3.57; *p* = 0.006)) after adjustment for many common risk factors, and the statistical association between stroke and MAC is a continuous variable: on multivariate analysis, each millimeter of calcification increased the relative risk of stroke by 1.24 (95 percent confidence interval, 1.12 to 1.37; *p* < 0.001) [70]. Nonetheless, the available evidence is not sufficient to draw valid conclusions about the prognostic weight of mitral valve calcifications in patients undergoing transcatheter mitral valve replacement.

It seems clear, in conclusion, that the association between stroke and mitral valve procedures is multifactorial. Many conditions, such as procedural complexity, myocardial or atrial tissue damage during grasping, air embolism, acute thrombus formation, paradoxical embolism through septal defect, mitral valve calcifications and mobilization of pre-existing thrombus (mainly in patients affected by atrial fibrillation) are potentially involved in peri- and post-procedural stroke [11]. These challenging clinical scenarios will be addressed in future prespecified randomized clinical trials in order to explore the epidemiological and pathophysiological association between stroke and mitral valve transcatheter interventions.

### 3.3. Differences in New Cerebral Lesions after TEER vs. TAVR Procedures

There appear to be some differences in the distribution pattern of the lesions between edge-to-edge transcatheter mitral valve repair and TAVR [28,71]. Both procedures are associated with the development of new ischemic lesions at brain MRI (85% in MitraClip and 80–85% in TAVR patients), but total lesion volume is reported to be higher in TAVR patients [11,57,71]. The distribution pattern is bilateral in both groups, suggesting an embolic origin of these lesions [72]. In the MitraClip population, most of the lesions at MRI were localized in the vascular territory of the middle and posterior cerebral artery; in patients undergoing TAVR procedure, the majority of embolic foci were found in the anterior and vertebrobasilar artery distribution territories, although these trends are not significant [57,71].

In TAVR patients, hyperlipidemia, renal dysfunction, lower aortic atheroma thickness, porcelain aorta, increased left atrial appendage velocity, and reduced aortic valve area at baseline were potentially associated with the number of new foci at brain MRI [71]. Another study involving 81 patients who underwent TAVR with a dual filter–based embolic protection device (Montage Dual Filter System, Santa Rosa, CA, USA) [73], identified balloon expandable transcatheter heart valves (THV) and post-procedural dilatation as independent predictors of new embolic lesions, suggesting that the mobilization of calcified tissue from the valve apparatus could be the main mechanism. On the other side, the origin of embolic particles is less clear in patients undergoing MitraClip.

### 3.4. Left Atrial Appendage Occlusion

Atrial fibrillation (AF) is the most common cardiac arrhythmia in the general population. Its actual estimated prevalence in adults is between 2 and 4%, and it is expected to be higher in the next decades, both in Europe and the U.S., because of population aging and intensifying search for undiagnosed AF [74,75]. AF increases the risk for ischemic stroke, silent cerebral ischemia and related cognitive dysfunction [76]. Prothrombotic processes due to blood stasis, mostly in the left atrial appendage (LAA), are responsible for atrial thrombus formation. This is particularly true in the setting of nonvalvular AF, as demonstrated by Blackshear and Odell [77], while in AF in patients with hemodynamically significant mitral stenosis and mechanical prosthetic heart valves the thrombogenic mechanisms are more complex [78]. Different clinical parameters are used for the stratification of embolic risk in AF, and the most important are summarized in the CHA_2_DS_2_-VASc score (which assigns 1 point for history of Congestive heart failure/left ventricular dysfunction, 1 point for Hypertension, 2 points for Age ≥ 75 years, 1 point for Diabetes Mellitus, 2 points for history of Stroke/transient ischemic attack/thromboembolic event, 1 point for Vascular disease (previous myocardial infarction or revascularization, peripheral artery disease, or aortic plaque), 1 point for Age 65–74 years and 1 point for female Sex). The anatomy of the LAA, which has been divided in four subtypes, appears also associated with thromboembolic risk, with the “chicken wing” configuration being associated with the lowest risk [79]. Amongst patients with an identified left atrial thrombus/spontaneous echo contrast, some predictors of thrombosis nonresolution after 3 weeks of anticoagulation have been identified: LAA emptying velocities < 20 cm/s at transesophageal echocardiography, an indexed left atrial volume > 40 mL/m^2^ and a multilobular LAA. [80]

Anticoagulation with Vitamin K Antagonists (VKAs) or Direct Oral Anticoagulants (DOACs), following annual stroke-risk evaluation with the CHA_2_DS_2_-VASc score, represents the standard care in patients with AF, in order to prevent stroke and peripheral embolization [75,81]. Although these drugs are widely used, and DOACs have shown non-inferiority–and sometimes even superiority–for stroke prevention while having significantly fewer bleeding complications compared to Warfarin [82,83,84,85], some issues still remain a matter of debate, including contraindications, side effects and adherence.

There is a small but significant percentage of patients who cannot tolerate oral anticoagulation because of high bleeding risk or major bleeding complications. These patients may benefit, as demonstrated in the past few years, from left atrial appendage occlusion (LAAO)/exclusion, with a transcutaneous or surgical approach, to reduce the risk of thrombotic embolism [75,86]. Various devices have been developed for percutaneous LAAO in the past years; not all of them have been approved in Europe, and some of them are still under clinical investigation [87]. Currently, the vast majority of implanted devices for catheter-based LAAO are the Watchman and the Amulet devices [88,89]. The procedure always requires transesophageal echocardiography, anesthesia, femoral vein access and transseptal puncture with a 14 French catheter system [90].

WATCHMAN LAA occluder: It is important to emphasize that this is the only device that has been tested in randomized clinical trials [88,91]. Experience with other devices is more limited and, consequently, less data are available.

While peri-procedural complications have been reported, including pericardial effusion, cardiac tamponade, device embolization and major bleeding, major stroke appears to be a rare peri-procedural event [92,93,94]. Many possible mechanisms can explain cerebral embolism during transcatheter closure of the LAA, such as air embolism during contrast dye injection or saline flush, embolism of parts of the interatrial septum during transseptal puncture, embolism of thrombi from guidewires and mobilization of thrombotic material from the LAA during catheter manipulation [90]. All these mechanisms can generate major clinical strokes (which are the only ones reported in clinical randomized trials and registries), but more often they can be associated with microembolism [95].

Two randomized clinical trials have tested safety and efficacy of the Watchman occluder: PROTECT AF (Watchman Left Atrial Appendage System for Embolic Protection in Patients With Atrial Fibrillation) and PREVAIL (Prospective Randomized Evaluation of the Watchman LAA Closure Device In Patients With Atrial Fibrillation Versus Long Term Warfarin Therapy), both designed to show non-inferiority against dose-adjusted Warfarin. In the PROTECT AF trial [94], peri-procedural clinical stroke incidence in 463 patients implanted was about 1%. This percentage has further decreased in the PREVAIL trial, where a 0.4% was reported. The decreasing trend has been confirmed also by the subsequent prospective registries (CAP [93], CAP2 [92], EWOLUTION [96]), which have shown no procedure-related strokes. All these data were obtained in patients treated with Warfarin for 45 days post implantation, followed by aspirin and clopidogrel for 6 months and then aspirin alone. The only exception was the EWOLUTION registry, where 27% of patients were treated with oral anticoagulants, 59% with dual antiplatelets, 7% with a single antiplatelet; 6% received no antithrombotic therapy. The cumulative post-approval success procedural rate was 95.6%, with peri-procedural stroke occurring in 0.08% of cases overall [96].

Majunke et al. [95] assessed cerebral embolic events, both clinical and silent ones, with (TCD) and MRI in a cohort of patients undergoing LAAO with Watchman device. Numerous microembolic signals were observed with periprocedural TCD monitoring; new asymptomatic ischemic cerebral lesions were observed with DW-MRI in 32% of patients soon after the procedure, and these were not related with the number of microembolic signals observed on TCD. Only a third of these ischemic lesions were still detectable in a second MRI performed 45 days later. These results suggest that transcatheter LAA closure is associated with asymptomatic cerebral embolism, even if most of the cerebral lesions are small and may disappear within 45 days [95].

AMPLATZER CARDIAC PLUG (ACP) and AMULET: Only case series and post-marketing observational registries are available for the ACP and Amulet devices. Patients were treated with variable combinations of aspirin and clopidogrel at the time of the procedure. Data from large multicenter ACP studies consisting of 1822 patients with an average follow up of 16 months showed a procedure-related stroke incidence between 0.8 and 1.2% [89,90,92]. The most recent data for the Amulet device come from a large prospective registry [97]. In 1078 implanted patients, 4 cases of peri-procedural stroke were reported. No data are available about microembolism and silent cerebral lesions. Only a few patients undergoing LAAO with the Amplatzer devices have been included in a small observational study, in which there was reported a low rate of new post-procedural SBIs [98].

LARIAT: This device (SentreHeart, Redwood, CA, USA) is characterized by percutaneous left atrial appendage suture ligation from dual-wire access with femoral and epicardial approach. As LAA seems to be a non-pulmonary vein focus of origin of AF, the epicardial-based exclusion procedure might also have the benefit of reducing AF burden. Two prospective [99,100] and one retrospective [101] registries are currently available on the device. The procedural success rate was between 94 and 98%. No peri-procedural stroke was observed. Nearly two-thirds (59%) of patients were treated with a single antiplatelet agent at the time of procedure, while 22% were on dual antiplatelet drugs and 19% on oral anticoagulation. No data are available about silent strokes and small brain lesions.

A few important studies are ongoing and can potentially change the utilization of LAAO and improve the knowledge of peri-procedural risk on the brain: The Amplatzer Amulet LAA Occluder trial (Amulet IDE, NCT02879448), a prospective randomized multicenter worldwide trial randomizing patients in a 1:1 fashion to either the Amulet device or the Watchman device; and the Interventional Left Atrial Appendage Closure vs. Novel Anticoagulation Agents in High-risk Patients with Atrial Fibrillation (PRAGUE-17 Study, NCT02426944). Both of them will evaluate, as a part of primary safety endpoint, the rate of peri-procedural brain lesions. Overall, more than 1200 patients will be enrolled.

In conclusion, LAAO represents a safe alternative for those patients who are not eligible for oral anticoagulation. Peri-procedural major stroke is a very rare complication. However, more data are needed about long-term outcome of patients with MRI-detectable brain lesions, as microembolism and silent ischemic strokes seem to be frequently associated with this procedure.

## 4. Conclusions

In the past decade, the use in clinical practice of noninvasive or less-invasive transcatheter cardiac procedures has expanded, with an increasing number of patients taking advantage of these new therapeutic strategies. Newer technologies are allowing indications for the use of these devices to change, extending the use to younger and lower surgical-risk patients, thus increasing the number of treated patients in the years to come. Some evidence suggests that cerebral embolism is underestimated and the rate of silent ischemic stroke or silent cerebral ischemia is much higher than the one reported in the available studies. Despite the lack of clinical manifestations, the cumulative effect of this cerebral ischemia may be linked to dementia and cognitive dysfunction. Based on these considerations, much attention should be drawn to cerebral embolism and preventive strategies to further reduce peri- and post-procedural risk of silent cerebral ischemia.

## Figures and Tables

**Figure 1 medicina-58-00045-f001:**
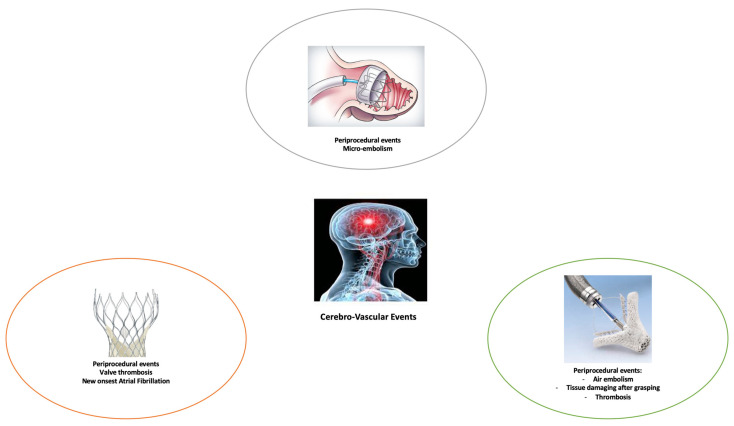
Mechanisms of cerebral ischemia in percutaneous non-coronary intervention procedures: transcatheter aortic valve implantation (TAVI) (orange); MitraClip (green); Left atrial appendage occlusion (grey).

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
