# Peer review of "Asymptomatic Stroke in the Setting of Percutaneous Non-Coronary Intervention Procedures"

_medicina, 2021, doi:10.3390/medicina58010045_

Round 1

Reviewer 1 Report

This is a review article, aimed to summarize the available data on “asymptomatic stroke in the setting of percutaneous non-coronary intervention procedures”.

The topic of the article, concerning the peri-procedural risk of silent ischemic cerebral lesions development is very important for clinical practice of both, cardiologists and neurologists. The autors have managed to present briefly the results of the main studies on transcatheter aortic valve replacement (TAVR), MitraClip procedure and Left atrial appendage occlusion (LAAO).

However, several points in this review might be improved:

  1. My suggestion is the text to be re-structured, for example for each interventional procedure to be discussed:

(1) What is known (trials results);

(2) The remained ambiguities and

(3) The main predictors of “asymptomatic” stroke, cognitive disorders and neuropsychological deficits to be highlighted.

  1. The following data could be presented in separate sections:
  • Different pattern of the lesions in MitraClip vs TAVR  
  • The role of protection devices

  1. The factors related to the left atrial appendage thrombosis (LAA empting velocity, number of LAA lobes, etc.) in AFib patients should be highlighted (as shown in the article of Naydenov et al, published in Medicina. 2021, 57:554-563)

Author Response

Point 1: My suggestion is the text to be re-structured, for example for each interventional procedure to be discussed:

(1) What is known (trials results);

(2) The remained ambiguities and

(3) The main predictors of “asymptomatic” stroke, cognitive disorders and neuropsychological deficits to be highlighted.

Response 1: thanks to reviewer 1 for this comment, we appreciate the suggestion.

The text is now re-restructerd following his indications. (pag 4, line 121-125 and 128-141; pag 6 line 314-324)

Point 2: The following data could be presented in separate sections:

  • Different pattern of the lesions in MitraClip vs TAVR
  • The role of protection devices

Response 2: thank the reviewer again for this comment. We have added new sections as suggested by the reviewer. (page 7 line 419-438)

Point 3: The factors related to the left atrial appendage thrombosis (LAA empting velocity, number of LAA lobes, etc.) in AFib patients should be highlighted (as shown in the article of Naydenov et al, published in Medicina. 2021, 57:554-563)

Response 3: factors related to LAAO have been highlighted and reference added. (page 8 line 458-462; pag 9 line 524-526; ref n. 80)

Reviewer 2 Report

In general, the revies by Giovanni Ciccarelli is very well written and very interesting. However, it unfortunately does not hold its promises: The title suggests that the review is about asymptomatic stroke in the setting of percutaneous non-coronary intervention procedures. However, in fact, most of the paper describes differences in symptomatic stroke. Furthermore, information about stroke linked to coronary procedures is described in the Introduction. Furthermore, some comments from the authors are directly in the text.

Author Response

Response to Reviewer 2 Comments

Point 1: In general, the revies by Giovanni Ciccarelli is very well written and very interesting. However, it unfortunately does not hold its promises: The title suggests that the review is about asymptomatic stroke in the setting of percutaneous non-coronary intervention procedures. However, in fact, most of the paper describes differences in symptomatic stroke. Furthermore, information about stroke linked to coronary procedures is described in the Introduction. Furthermore, some comments from the authors are directly in the text.

Response 1: thanks to reviewer 2 for this comment, we appreciate the suggestion.

The text is now re-restructerd following his indications. In particular, the paper is now more focus on silent cerebral ischemia and some sentences regarding percutaneous coronary intervention have been removed; the introduction describes the role of coronary tools in the pathogenesis of SIB in the setting of non-coronary percutaneous intervention (page 1 and 2); the comments have been removed.
